# Surface-Enhanced Raman Spectroscopy (SERS) Investigation of a 3D Plasmonic Architecture Utilizing Ag Nanoparticles-Embedded Functionalized Carbon Nanowall

**DOI:** 10.3390/nano13192617

**Published:** 2023-09-22

**Authors:** Chulsoo Kim, Byungyou Hong, Wonseok Choi

**Affiliations:** 1Department of Electrical Engineering, Hanbat National University, Daejeon 34158, Republic of Korea; msdkcs1@gmail.com; 2School of Electronic and Electrical Engineering, Sungkyunkwan University, Suwon 16419, Republic of Korea; byhong@skku.edu

**Keywords:** surface-enhanced Raman scattering (SERS), plasmonic nanoparticles (PNPs), carbon nanowall (CNW), silver nanoparticles (Ag NPs), rhodamine 6G (R6G)

## Abstract

Surface-enhanced Raman scattering (SERS) is a highly sensitive technique for detecting DNA, proteins, and single molecules. The design of SERS substrates plays a crucial role, with the density of hotspots being a key factor in enhancing Raman spectra. In this study, we employed carbon nanowall (CNW) as the nanostructure and embedded plasmonic nanoparticles (PNPs) to increase hotspot density, resulting in robust Raman signals. To enhance the CNW’s performance, we functionalized it via oxygen plasma and embedded silver nanoparticles (Ag NPs). The authors evaluated the substrate using rhodamine 6G (R6G) as a model target molecule, ranging in concentration from 10^−6^ M to 10^−10^ M for a 4 min exposure. Our analysis confirmed a proportional increase in Raman signal intensity with an increase in concentration. The CNW’s large specific surface area and graphene domains provide dense hotspots and high charge mobility, respectively, contributing to both the electromagnetic mechanism (EM) and the chemical mechanism (CM) of SERS.

## 1. Introduction

The phenomenon of surface-enhanced Raman scattering (SERS), initially by authors including Fleischmann in 1974, has undergone extensive study over the past 5 decades [1]. It has found applications in various fields, including bio-analysis, chemical analysis, and biomedical sciences. Traditional methods like enzyme-linked immunosorbent assay (ELISA), high-performance liquid chromatography (HPLC), and isotope dilution mass spectrometry (IDMS), while reliable, are often expensive and time consuming. However, the current era demands high-sensitivity, cost-effective, and rapid analytical methods. SERS, a label-free detection method known for its high sensitivity, resolution, affordability, and user-friendly interface, has emerged as a promising solution for bioanalytical applications. This technique enables chemical fingerprinting through a specific spectral signature, making it suitable for analytical methods requiring ultra-trace detection limits in parts per million (ppm) and parts per billion (ppb) units [2]. SERS operates through two mechanisms: the electromagnetic mechanism (EM) and the chemical mechanism (CM) [3]. EM is attributed to the phenomenon of local surface plasmon resonance, which occurs when specific metal nanostructures like plasmonic nanoparticles, nanospheres, and nanoshells are exposed to light of a specific frequency [4,5]. Incident photons induce vibrations of free electrons on the surface of these nanostructures, generating a local electromagnetic field known as a “hot spot”. In contrast, CM is primarily related to the electron arrangement at the interface between the detection material and plasmonic nanoparticles, involving electron transitions between the highest occupied molecular orbital (HOMO) and the lowest unoccupied molecular orbital (LUMO). However, the influence of CM in SERS is relatively limited [6], with EM being the dominant factor. Given the critical role of providing high-density hotspots in SERS, researchers have explored three-dimensional nano-architectures (3D NAs) like nanopyramids and metal–organic frameworks. However, the fabrication of 3D NAs often involves complex processes such as imprinting and lithography, and it may rely on specific materials with limitations in terms of mechanical properties and reproducibility. In this study, we propose the use of functionalized carbon nanowall (FCNW) as a scaffold for silver nanoparticles (Ag NPs) in a 3D NA structure. FCNWs can be synthesized in a one-step process using plasma-enhanced chemical vapor deposition (PECVD) [7]. The pristine material, carbon nanowall (CNW), consists of vertically aligned graphene domains and has demonstrated excellent performance in various applications, including gas sensors and field-effect transistors, due to its high electron mobility, mechanical strength, and chemical stability [8,9,10]. In a previous study, the synthesis of CNW and Ag NPs in a 3D lamellar structure showed promise as a SERS substrate. Although the plasmons of graphene domains in CNWs occur in the terahertz frequency range, they do not directly contribute to localized surface plasmon resonance (LSPR) enhancement. However, they provide high-density hot spots due to the large specific surface area of CNW [11]. Furthermore, as reported in the literature, graphene and graphene oxide can enhance the CM, and the high electron mobility of CNW contributes to a low CM ratio [12]. The combined enhancement of EM and CM has a synergistic effect on SERS improvement. Compared to CNW, FCNW offers significant enhancements in CNW-metal/molecule bonding, and the introduction of oxygen groups through plasma functionalization results in more stable SERS substrates. In this study, we successfully collected a sensitive SERS signal for rhodamine 6G (R6G) using the proposed Ag NP-embedded FCNW substrate. This study demonstrates the potential of Ag NP-embedded FCNWs as a SERS substrate for sensitive in situ detection of R6G.

## 2. Materials and Methods

### 2.1. Reagents and Materials

Trichloroethylene (EP grade) for wafer cleaning was purchased from Junsei Chemical Co., Ltd. Methanol (Tokyo, Japan) (ACS grade) and acetone (ACS grade) was purchased from Honeywell International, Inc. (Mecklenburg County, NC, USA) and used without further purification. R6G, used as a model target molecule, was obtained from Sigma-Aldrich (St. Louis, MO, USA).

### 2.2. Fabrication of Ag Nanoparticles–Embedded Functionalized Carbon Nanowall

To use CNW as a SERS substrate, first, CNW was grown on a Si substrate by microwave plasma-enhanced chemical vapor deposition (ASTeX-type, MPECVD, Woosin CryoVac (Uiwang-si, Republic of Korea), 2.45 GHz microwave). Hydrogen gas (purity 99.9999%) was first injected into the chamber to form a hydrogen atmosphere, and then methane gas (purity 99.999%), a precursor of CNW, was injected. The gas ratio in the chamber was H_2_:CH_4_ = 40:40 sccm until 90 s (step of the carbon particulate formation and nanographene sheets generation) into the process, and then H_2_:CH_4_ = 40:20 sccm. Manipulation of these gas ratios is an important factor in determining the CNW growth rate and graphitization properties. After CNW growth, it was cooled to room temperature at a high vacuum (3 × 10^−2^ Torr). Secondly, functionalization was performed while maintaining a vacuum in the chamber. This is an in situ process for the synthesis and functionalization of CNW. A plasma ball was formed at 300 °C with 500 W of microwave power while injecting 20 sccm of oxygen, which lasted for 20 s. In this case, oxygen radicals can cause destruction or etching of CNW, and the process time is important because a short time reduces the CNW branch removal rate. In the functionalization step, oxygen radicals remove the carbon branches present in the CNW and simultaneously impart functional groups to the graphene domains, thereby increasing the deposition rate of Ag NPs. The embedding of Ag NPs into the FCNW was performed via a radio frequency (RF) magnetron sputtering system (ITS, PG600A600W, Daejeon, Republic of Korea). At this time, 80 sccm of argon gas was injected, and a Ag target (purity of 99.999%) was used. To help understand the process, a schematic diagram of the process is shown in Figure 1. R6G was adopted as a model target molecule for SERS signal detection to evaluate the performance of the Ag NPs-embedded FCNW substrate. SERS substrates are prepared by soaking in a solution of R6G molecules (10^−6^–10^−10^ M) for 1–5 min, followed by final drying with a nitrogen injection.

### 2.3. Characterization and Analysis of Materials

All Raman spectra, including the SERS analysis, were analyzed using Raman spectroscopy (HORIBA, Osaka, Japan, LabRAM HR-800). The laser power was 3 mW, the excitation wavelength was 532 nm, and a × 50 objective with NA = 0.5 was used. The morphological characteristics were analyzed by using a field-emission scanning electron microscopy (FE-SEM, HITACHI, Tokyo, Japan, S-4800) at 15 kV. Surface activation energy analyses were performed by using a water contact angle (WCA) analyzer (SEO, Phoenix MT, Seoul, Republic of Korea). Characterization of the pore distribution of the samples was performed using mercury intrusion porosimetry (Micromeritics, Norcross, GA, USA, AutoPore 9520).

## 3. Results and Discussion

### 3.1. Raman Shift of CNW and FCNW

The carbon molecules that constituted the graphene domains show the Raman active bands in the spectrum. Representatively, there are D-peak, G-peak, and 2D-peak near 1350, 1570, and 2700 cm^−1^, respectively [13]. The D-peak is due to out-of-plane vibrations at defects in carbon atoms, whereas the G-peak is due to in-plane vibrations attributed to sp^2^ bonding in carbon [14]. Figure 1a,c show the Raman spectrum of a pristine CNW and the I_D_/I_G_ ratio, respectively. CNW has many edges, a high D-peak is observed, and a sharp G-peak is observed because it is a material composed of sp^2^ combined carbon atoms [13,14,15]. The 2D-peak can also be attributed to the double resonance of carbon atoms and is also observed in CNW, confirming that CNW has a graphene-based 3D architecture [16]. In this paper, the growth time varied from 300 to 900 s. Among them, the sharpest 2D-peak was observed at 600 s, suggesting that the graphene sheets with the fewest number of layers were vertically oriented [17,18]. In the Raman spectrum, FCNW increased the ratio of D-peak, which may be due to the modification of the hexagonal carbon ring by oxygen radicals or the multiple of functional groups. However, the highest 2D-peak was also observed for the 600 s sample. Figure 2 shows that the scenarios in (b) and (d) can also be known through the I_2D_/I_G_ ratio [19]. Disorder and defects in carbon materials can be evaluated easily through the I_D_/I_G_ ratio, but they are not completely reliable considering the ambiguity of the analysis due to the overlapping of the D-peak and G-peak.

### 3.2. Morphological Characterization for Plasmonic Nanoparticles

CNWs have a bottom layer of approximately 20 or more layers of graphene, which grows vertically at the interface of localized graphene islands [20]. In our study, if nanographene is formed about 90 s after the start of the process and the methane gas ratio is maintained at 40 sccm, it may be advantageous in terms of the growth rate, but the yield of amorphous carbon branches also increases. To reduce the yield of carbon branches, the methane gas flow rate was reduced by half after 90 s of the process, but it was not possible to completely control the growth of carbon branches. Figure 2a shows FE-SEM images of the pristine CNW surface. The amorphous carbon branches, consisting of sp^3^ bonds, significantly contribute to the increase in the D-peak in the Raman spectrum, and, in this study, they were removed in the functionalization step using oxygen plasma [21]. It should be noted that during the functionalization step, destruction of the nanowall occurred due to the saturation of the oxygen functional groups. Figure 2b shows that physical destruction of the nanowall did not occur, round edges were maintained, and FCNWs with carbon branches removed were observed. The WCA of the CNW was 62.7 degrees, whereas the FCNW decreased to 23.4 degrees (Figure 2c,f). The formation of functional groups by oxygen plasma generates a Laplace pressure in the same direction as gravity due to an increase in surface energy, thereby lowering the surface WCA [22,23,24]. Functional groups present on the FCNW basal plane and edge create sites for potential interactions with Ag NPs. The charged functional groups of the FCNW can cause electrostatic interactions or coordination bonds with Ag NPs, which can consequently increase the rate of Ag NPs-embedding. The functional group is not directly involved in SERS. Ag NPs were embedded into the FCNW by RF-magnetron sputtering, and the corresponding images are shown in Figure 2g. The increase in FCNW edge thickness is due to the insertion of Ag NPs, which can induce changes in surface energy. The corresponding schematic illustration and WCA analysis results are included in Figure 2h,i, respectively.

The surface morphological characteristics of CNW and FCNW are differentiated as shown in Figure 3a, due to the removal of amorphous carbon branches by the oxygen plasma ball in the functionalization step. Based on the interface, the left side is the FCNW area and the right side is the CNW area. CNW has a variety of edge shapes. In previous studies, shapes and patterns such as zigzags and rounds were found [25,26]. Although not reported in the literature, the round shape contains the largest pores and is suitable for nanoparticle embedding. The porosity distribution can be seen in Figure 3(b-1–b-3). Pores exist on the surface of the CNW and the FCNW, and the porosity distribution is an essential factor when embedding nanoparticles. The pore distribution of 100 to 200 nm is dominant, but sometimes the pore diameter is smaller than 100 nm or larger than 200 nm. Pores with a diameter of less than 100 nm are predominantly deposited only on CNW edges during Ag NPs-embedding using sputtering. In addition, Ag NPs-embedding in pores with a diameter exceeding 200 nm has a high probability of being inserted between pores to form a thin film. In this paper, the authors have determined that a pore diameter within the range of 100 to 200 nm is optimal for sputtering. Looking at the pore diameter distribution in Figure 3c, it was confirmed that the overall diameter increased, which was due to the removal of carbon branches. The decrease in pore diameter in Figure 3(b-3) is due to the insertion of Ag NPs into the FCNW edge according to the distribution of pore diameters. Nevertheless, it can be seen from the Gaussian fitting curves in Figure 3(b-1–b-3) that pores with an average diameter of 100–200 nm dominate, as the authors intended.

### 3.3. SERS Activation on the Ag NPs-Embedded FCNW

Figure 4a shows the molecular formula of R6G and the Raman spectrum of pristine CNW exposed to high concentrations of R6G. SERS substrates using a pristine CNW are capable of identifying high concentrations of R6G, but they exhibit low intensity. This can be inferred to be due to the slight contribution of CM due to the high electron affinity [27], which is a physical property of CNW, and molecular detection at low concentrations is limited. The Ag NPs-embedded FCNW substrate shows an exposure time response of 1–5 min with R6G used as a model target molecule at a certain concentration (10^−7^ M), as shown in Figure 4b. The peak around 616 cm^−1^ is due to C-C-C ring in-plane vibrations, and the peak at 778 cm^−1^ is due to C-H out-of-plane vibrations [28,29]. In addition, the peaks observed at near 1365, 1514, 1577, and 1654 cm^−1^ are aromatic stretching modes present in the R6G molecule, and they are considered representative Raman spectra of R6G [30]. The intensity of the peaks decreased according to the increased exposure time described in Figure 4b, because the dynamic behavior of R6G molecules adsorbed on the surface of Ag NPs was dependent on the exposure time [31].

Raman mapping is a robust tool for visualizing the heterogeneity in intensity differences across substrates. Figure 5c illustrates the Raman spectral mapping images corresponding to various substrates. The exposure time for all experiments was fixed at 4 min with a concentration of a 10^−7^ M R6G aqueous solution. Raman mapping images obtained from pristine CNW exhibit high localization and limited intensity, potentially explaining the remarkably low R6G detection limit. This observation is further supported by the Raman spectrum in Figure 5a. And Table 1 shows samples corresponding to Figure 5a. Although pristine CNWs do not directly affect the detection of model target molecules, they can be used as architectures with a large specific surface area for providing a high density of SERS active sites. Conversely, CNW embedded with Ag NPs, despite having a low density, exhibits Raman mapping intensity capable of identifying R6G. This phenomenon can be attributed to the high plasmonic activity of Ag NPs embedded within the extensive specific surface area of CNW. Raman mapping images of Ag NPs-embedded FCNW substrates display exceptionally high intensity. The presence of oxygen functional groups in FCNW contributes significantly to increased Ag NPs embedding yield through enhanced van der Waals bonds, hydrogen bonds, or other interactions with molecules [32,33,34,35]. The resulting high-density plasmons, stemming from the increased Ag NPs embedding, emerge as the dominant factor responsible for SERS enhancement. A schematic illustration to aid reader comprehension is included in Figure 5b. Ag NPs-embedded FCNW (samples 1–5) showed very strong SERS signals. Non-functionalized CNW (samples 6–10) with embedded Ag NPs showed lower peaks than Ag NPs-embedded FCNW. It has been confirmed that carbon branches are considered to be a very limiting factor for the embedding of Ag NPs. Pristine CNW (samples 11–15) display minimal peaks, with the target molecule remaining unidentified. This further suggests that the contribution of CNW’s CM is minimal, aligning with the findings from Raman mapping. The SERS enhancement factor (EF) was calculated using Equation (1) [36]:(1)$$EF=\frac{I_{SERS}/N_{SERS}}{I_{REF}/N_{REF}}$$

Here, *I_SERS_* and *I_REF_* represent the intensity in the Raman spectrum at 1365 cm^−1^ for R6G adsorbed on SERS and reference substrate, respectively. *N_SERS_* and *N_REF_* denote the number of R6G molecules absorbed on SERS and reference substrates, respectively. The SERS EF of Ag NPs-embedded FCNW was determined to be 6.817 × 10^7^. These findings highlight that, when compared to CNW and Ag NPs CNW, Ag NPs-embedded FCNW forms high-density plasmons, significantly influencing electromagnetic (EM) enhancement and contributing to the SERS signal enhancement.

## 4. Conclusions

Incorporating 3D nanoarchitectures (3D NA) with plasmonic nanoparticles (NPs) has been shown to yield high-level surface-enhanced Raman scattering (SERS) signals, making it a promising SERS substrate. The integration of nanoparticles into complex NA structures, such as carbon nanowall (CNW), demands advanced processing techniques. Nonetheless, our successful fabrication of Ag NPs-embedded FCNW using oxygen plasma underscores the viability of this approach. Our study demonstrates that the combination of FCNW and Ag NPs exhibits two synergistic effects as SERS substrates. Firstly, plasmonic NPs can be densely deposited on the ample specific surface area of FCNW, resulting in the distribution of dense hotspots. This, in turn, significantly contributes to electromagnetic (EM) enhancement by promoting more plasmon formation. Secondly, the binding of Ag NPs to R6G, a model target molecule, facilitates charge transfer with high probability, making a slight but notable contribution to chemical (CM) enhancement. In conclusion, the Raman peak of R6G exhibited a robust SERS activity signal on the Ag NPs-embedded FCNW substrate. This innovative approach holds great potential for applications in the fields of bioanalysis and chemical analysis, paving the way for further SERS proof-of-concept research.

## Data Availability

The data that support the findings of this study are available from the corresponding author upon reasonable request.

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
