# Peer review of "Surface-Enhanced Raman Spectroscopy (SERS) Investigation of a 3D Plasmonic Architecture Utilizing Ag Nanoparticles-Embedded Functionalized Carbon Nanowall"

_nanomaterials, 2023, doi:10.3390/nano13192617_

Round 1

Reviewer 1 Report

In this review, Chulsoo Kim et al., reported a plasma-enhance chemical vapor deposition combined Ag nanoparticle deposition to formed the Ag-FCNW 3D structure for SERS measurement. Overall, this work is quite concise and reasonable, several minor concerns shall be addressed before publication:

1.      A main concern is related with the method:

a.       Page 2 line 89-91, the authors mentioned that the gas ratio is an important factor in CNW growth, please interpret more information for the ratio variation of CH4 from 40 sccm for 90 s at initial stage, and this value is decreased to 20 sccm in subsequent growth.

b.      Page 3 line 97, any interpretation for the plasma treatment duration of 20 s for final result.

2.      CNW sample with a growth time of 600 s reveal a sudden D peak intensity or defect increase in Figure 1, looks quite weird, any reasonable interpretation?

3.      Page 3 line 131-136, the interpretation for graphene Raman peaks is a bit confusion. Both half width at half maxima (HWHM) of 2D peak and I2D/IG can determine the thickness of intact graphene. (10.1103/PhysRevLett.97.187401) However, the HWHM at 600 s sample is contradict with the I2D/IG ratio in determining the graphene thickness. In defective graphene or chemical functionalized graphene (10.1021/jacs.6b02209), this rule is even complicated, please clarify this point.

4.      Page 5 line 179 and Figure 2. How the authors count the pore size for the irregular CNW, more detailed shall be supplemented.

5.      Following last concern, page 5 line 179, why pore size beyond the range of 100-200 nm reveals negative influence toward Ag NPs embedding? Any interpretation or reference.

6.      Figure 5a is wrongly presented.

7.      Figure 5c, why not choose a patterned CNW sample like that in Figure 3a to reveal the SERS intensity difference.

8.      Page 2 line 59, what dose LSPR mean, please denote this abbreviation.

Author Response

The author submits a response to your comment.

in addition the revised manuscript file is attached.

I have revised the thesis to reflect your opinions. I would like to thanks the reviewer who pointed out my paper.

Reviewer 2 Report

The development and application of new SERS substrates is a current trend, and the manuscript of C. Kim et al is contributing to this area. The authors proposed a new version of SERS substrate preparation using grown and functionalized carbon nanowalls, which were then coated with Ag. The results presented in the manuscript are novel and generally worthy of publication. However, I am against publication of the manuscript in its present form. There are two main problems with this manuscript:

1). The language and style of the text needs significant improvement. I even had trouble understanding the meaning of some sentences and statements; in some cases, the information from the text did not match the figures. Examples of the problematic text are in lines 17, 122, 149-152,169-172, 180-183, 190-192, 216-218, 240-241. In some cases, like the sentence “Pores with a diameter of less than 100 nm and greater than 200 nm negatively affect the uniformity of Ag NPs embedding.” (lines 179-180), the origin and validity of such statements is unclear to readers. There are also many typographical errors in the manuscript, including errors in the table in Fig. 5a-1.

2). The central goal of the manuscript is the SERS substrate. There is a standard approach to the study of new SERS substrates, which includes the Raman spectrum of the substrate without target molecules, the non-resonant spectrum of target molecules, and the SERS spectrum from the tested substrate with a known number of target molecules deposed. The authors of the manuscript did not show Raman spectrum from the empty Ag NPs-embedded FCNW, did not explain why the Raman peaks of CNW (Fig. 1a) disappear in Fig. 4a (CNW+R6G) and in corresponding spectra of Fig. 5a. Also, the protocol of the R6G deposition was not described in the manuscript. The variations in the deposition time are ambiguously related to the amount of analyte on the substrate. Thus, it is not clear whether the improved R6G detection for Ag NPs-embedded FCNW is associated with improved R6G adsorption or with increased SERS-effect on the substrate.

Author Response

(The authors gave the same response as above.)

Reviewer 3 Report

In this study, Kim et al. demonstrated the SERS-based detection of R6G molecules by using AgNPs embedded FCNW. The study is interesting with reasonable results, however, the manuscript cannot be accepted in its present form as there are a few issues associated with it. Hence, a major revision is required before acceptance. The must address the following concerns properly:

1) At present, the introduction is very weak. There should be more discussion on other noble metal nanoparticles as far as the SPR/LSPR is concerned. Similarly, there should be a brief discussion on nanostructured carbon materials, graphene, and CDs on textured Si substrates as they have been investigated for SERS detection including R6G molecules and the obtained results were excellent. The authors must provide some relevant discussion on graphene and CNW/VGNWs with proper references. The authors must refer to the following papers: https://onlinelibrary.wiley.com/doi/abs/10.1002/admi.202100977; https://www.sciencedirect.com/science/article/abs/pii/S016943322033244X

2) The authors have missed out on some recently (2022 and 2023) published articles on CNW/VGNWs, where IC-PECVD, r-PECVD, PEALD, etc., techniques have been used to produce CNW/VGNWs on different substrates at different temperatures, and new insights have been presented regarding the growth mechanism. Moreover, morphology and Raman analyses of different samples require detailed analyses as they are the backbone of the present study. The authors must go through these important articles: https://pubs.acs.org/doi/abs/10.1021/acsaelm.1c00807; https://pubs.acs.org/doi/full/10.1021/acsami.1c21640; https://pubs.acs.org/doi/abs/10.1021/acsanm.2c03006; https://pubs.acs.org/doi/10.1021/acsami.0c19188

3) Please include the TEM analysis of the AgNPs embedded in FCNW. Is the porosity distribution histogram of Ag NPs–embedded FCNW correspond to Fig.2i & j? DI water is used for contact angle measurement? Then it should be water contact angle (WCA), not surface contact angle. Please modify suitably.

4) What is the average particle size of AgNPs? Does the particle size play any role in SERS detection? And how stable are they at ambient conditions since their chemical stability is very poor? So, the durability of such SERS substrates becomes important here. If possible please include the durability and repeatability tests.

5) A table of comparison should be included in order to have a better idea of the obtained results of SERS-based detection of R6G.

6) Did you carry out an interference (in the presence of other organic molecules) study of SERS-based detection? I think it would be interesting to see this. Moreover, please calculate the SERS enhancement factor (EF) as it is very much important in this case.

7) The title should be rephrased as it is confusing at present.

The English language requires minor editing including title, Fig captions, etc.

Author Response

(The authors gave the same response as above.)

Round 2

Reviewer 2 Report

In the revised manuscript, the authors significantly improved the text and presentation. The manuscript may be published as is, although I still believe that my suggestions regarding the presentation of the SERS research and quantification of the amount of adsorbed R6G molecules could be useful for readers. In the present version of the manuscript, I have difficulty understanding the sentence in lines 197-199; and lines 221-225 apparently contain two sentences with similar meaining.

Author Response

(The authors gave the same response as above.)

Reviewer 3 Report

The revised manuscript is now in better shape to be accepted for publication.

Minor English editing can be done during proof preparation.

Author Response

(The authors gave the same response as above.)
